EcoNicheS: enhancing ecological niche modeling, niche overlap and connectivity analysis using the shiny dashboard and R package

http://orcid.org/0000-0003-4685-5322 Sunny Armando 1 2 sunny.biologia@gmail.com
Marmolejo Clere 1 2
Vidal-López Rodrigo 2
http://orcid.org/0000-0002-9864-567X Falconi-Briones Fredy A. 3
http://orcid.org/0000-0002-8490-4095 Cuervo-Robayo Ángela P. 4 5
Bolom-Huet René 1 renblht@gmail.com
1 Centro de Investigación en Ciencias Biológicas Aplicadas, Facultad de Ciencias, Universidad Autónoma del Estado de México , Toluca, Estado de México , Mexico
2 Centro de Innovación Digital “Mandra” Laboratorio Nacional de Enseñanza e Innovación aplicando Cómputo de Alto Rendimiento (EICAR), CONAHCyT, Universidad Autónoma del Estado de México , Toluca, Estado de México , Mexico
3 Departamento de Conservación de la Biodiversidad, El Colegio de la Frontera Sur , San Cristóbal de Las Casas, Chiapas , Mexico
4 Laboratorio Nacional Conahcyt de Biología del Cambio Climático, CONAHCyT , Mexico City , Mexico
5 Departamento de Zoología, Instituto de Biología, Universidad Nacional Autónoma de México , Mexico City , Mexico
Steinke Dirk
Electronic publication date: 2025 Mar 28
Publication date: 2025
Volume: 13
Electronic Location ID: e19136
Received 2024 Sep 20; Accepted 2025 Feb 19
Copyright: © 2025 Sunny et al.
Copyright year: 2025
Copyright holder: Sunny et al.
License: This is an open access article distributed under the terms of the Creative Commons Attribution License, which permits unrestricted use, distribution, reproduction and adaptation in any medium and for any purpose provided that it is properly attributed. For attribution, the original author(s), title, publication source (PeerJ) and either DOI or URL of the article must be cited.
License URL: https://creativecommons.org/licenses/by/4.0/

Keywords: Ecological niche modeling, Shiny dashboard, R package, Habitat suitability, Species distribution modeling, Landscape connectivity, Niche overlap, Graphical user interface (GUI)

Funding: Secretary of Research and Advanced Studies (SYEA) of the Universidad Autónoma del Estado de México 4732/2019CIB, 6801/2022CID and 7194/2025CIB SEP PRODEP 511/2022-5401 CONACYT FOINS 6828/2017 This work was supported by the Secretary of Research and Advanced Studies (SYEA) of the Universidad Autónoma del Estado de México (grants to AS: 4732/2019CIB, 6801/2022CID and 7194/2025CIB), SEP (grant to AS: PRODEP 511/2022-5401) and CONACYT (grant to AS: FOINS 6828/2017). The funders had no role in study design, data collection and analysis, decision to publish, or preparation of the manuscript.

==============================
EcoNicheS (https://github.com/armandosunny/EcoNicheS) is a comprehensive R package built on a Shiny dashboard that offers an intuitive and streamlined workflow for creating ecological niche models (ENMs) and landscape connectivity models. It incorporates tools for niche modeling, overlap analysis, and connectivity modeling, leveraging robust algorithms from the biomod2 suite. EcoNicheS is designed to simplify the technical complexities of ENMs, bridging the gap between advanced modeling techniques and user accessibility. The package offers an interactive interface for streamlined data input, model parameterization, and result visualization. Its comprehensive toolset includes occurrence data processing, pseudoabsence point generation, urbanization filters, and ecological connectivity modeling, distinguishing it from other platforms. EcoNicheS integrates innovative workflows with dynamic output visualizations while emphasizing reproducibility and comparability across statistical methods. Its practical applications span diverse research fields, including biogeography, epidemiology, evolutionary studies, climate change impacts, landscape connectivity, and biodiversity conservation. This versatility makes EcoNicheS a valuable resource for advancing in ecological and conservation science.

Introduction

Portions of this text were previously published as part of a preprint (Sunny et al., 2024a).

Ecological niche modelling (ENM) can be used to estimate specie’s Grinnellian niche based on observed occurrence data and associated environmental variables (Peterson, 2011). Moreover, because ENM can be used for predicting potential geographic distributions under varying environmental conditions, they operate under the assumption that observed distributions reflect environments suitable for the species (Peterson, 2011; Araújo & Peterson, 2012; Franklin, 2023; Sunny et al., 2024a). Therefore, these models have become essential tools across various disciplines, including biogeography, conservation planning, climate change adaptation, and disease ecology (Araújo et al., 2019; Sillero et al., 2021; Espindola et al., 2022; Sunny et al., 2024a, 2024b; Franklin, 2023).

The use of ENM methodologies has continuously evolved, expanding beyond only estimating species distributions (Sunny et al., 2024a). These include reconstructing historical species distributions, ecological niche model comparison, assessing future biodiversity shifts due to global change, and guiding conservation strategies, invasive species management, and landscape connectivity planning (Peterson, 2011; Culshaw, Mairal & Sanmartín, 2021; Santini et al., 2021; Ahmed et al., 2022). ENM allows researchers to analyze species niche at an environmental or geographic dimension and across both spatial and temporal scales, providing critical insights into the ecological processes and biogeographical mechanisms shaping biodiversity patterns over time and across landscapes (Araújo et al., 2019; Sunny et al., 2024a). The versatility of these methods makes them valuable tools for addressing complex ecological questions and supporting decision-making processes to preserve biodiversity in face rapid environmental change (Koerich et al., 2020; Rubio-Blanco et al., 2024).

Despite their utility, ENM workflows are often technically complex, requiring proficiency in programming languages such as R or Python and the use of multiple specialized packages for data preprocessing, model calibration, validation, and visualization (Di Cola et al., 2017; Cobos et al., 2019; Kass et al., 2021; Sunny et al., 2024a). This complexity limits accessibility, particularly for conservation practitioners and decision makers who may lack coding expertise but require robust, actionable outputs (Sillero et al., 2023; Sunny et al., 2024a). To address these barriers, intuitive software platforms are needed to streamline ENM workflows, making them accessible without compromising analytical rigor (Jia et al., 2022; Sunny et al., 2024a).

To meet this need, we introduce EcoNicheS, an innovative R package built on the ShinyDashboard framework, which provides a user-friendly interface for ecological niche modeling and functional connectivity analysis. Shiny enables users to develop responsive interfaces that leverage R’s computational capabilities while maintaining intuitive, browser-accessible designs (Fig. 1, Chang et al., 2025). Its structured layouts and dynamic dashboards are well-suited for modular workflows, allowing seamless integration of multiple analyses. EcoNicheS simplifies complex ENM processes and complements existing tools such as Wallace 2 and ntbox (Osorio-Olvera et al., 2020; Kass et al., 2023). It integrates key functionalities, including data preprocessing, ensemble modeling, niche overlap analysis, and functionally connectivity modeling, ensuring a streamlined experience for users of all expertise levels (Araújo et al., 2019; Zurell et al., 2020; Kass et al., 2023). By following best practices in the ENM and emphasizing reproducibility, GIS compatibility, and interactive visualization, EcoNicheS enhances accessibility while maintaining analytical rigor, facilitating actionable insights for researchers and practitioners alike (Osorio-Olvera et al., 2020; Sunny et al., 2024a).

Figure 1 User interface of EcoNicheS, showing the different modules that comprise the platform.

(1) Environmental data, (2) retrieve and clean GBIF data, (3) load and plot maps, (4) correlation layers, (5) points and pseudoabsences, (6) biomod2, (7) load and plot maps (post-modeling), (8) partial ROC analysis, (9) remove urbanization, (10) calculate area, (11) gains and losses plot, (12) niche overlap (via ENMTools), (13) map inverter, (14) functional connectivity (Circuit theory), (15) functional connectivity (Least-cost path, LCP).

In this study, we present EcoNicheS, detailing its features, installation, and use. To demonstrate its practical utility, we conducted a case study modeling the ecological niche and functional connectivity of Tapirus bairdii (the Central American tapir) in the Selva Maya, a tropical rainforest spanning Mexico, Belize and Guatemala (Sunny et al., 2024a). This case study illustrates how EcoNicheS can generate meaningful insights for species conservation and management (Sunny et al., 2024a), highlighting its potential as a powerful tool for biodiversity research and decision-making.

Materials and Methods

Information included in each module

EcoNicheS, developed using the Shiny Dashboard framework, is an open-access platform design to provide an intuitive interface for creating ecological niche models. By streamlining the modeling process, EcoNicheS simplifies the inherent complexity of each step. The workflow is structured into 15 modules (Fig. 1): Occurrence Processing, Load and Plot Maps (performed twice); Correlation Layers, Points and Pseudo-Absences; ENM using the biomod2 library; Partial ROC Analysis; Removal of Urbanization Effects; Calculation of Area and Gains/Losses; Niche Overlap Analysis via ENMTools; and Connectivity Analysis (Figs. 1, 2, Table 1). The folder structure and outputs are detailed in Fig. 3. A user manual, including step-by-step screenshots of each module and their results, is available alongside the package code and installation instructions at https://github.com/armandosunny/EcoNicheS, (Sunny et al., 2024a).

Figure 2 Workflow of the EcoNicheS application.

The sequential steps and modules (represented by gray circles) within the platform, along with the specific procedures executed at each stage.

Table 1 Overview of each module’s analyses, including relevant references and the corresponding R packages used.

Module	Module name	Analyses	Packages	References	
Module 1	Environmental data	Download and process WorldClim bioclimatic layers, including bioclimatic variables (Bio), minimum temperature (Tmin), maximum temperature (Tmax), average temperature (Tavg), wind speed (Wind), and vapor pressure (Vapr).	geodata, terra, countrycode	Hijmans (2024a, 2024b), Arel-Bundock, Enevoldsen & Yetman (2018)	
Module 2	Occurrence processing	Obtain and process occurrence records of the target species, identify and correct database errors, and minimize sampling bias.	rgbif, user data, rgeos, CoordinateCleaner, countrycode, SpThin	Aiello-Lammens et al. (2015), Arel-Bundock, Enevoldsen & Yetman (2018), Zizka et al. (2019), Bivand & Rundel (2023), Chamberlain et al. (2024)	
Module 3	Load and plot maps	Load and visualize a .asc file of the study area, with options to view and save the image in .pdf format.	leaflet, leaflet.extras, raster	Cheng et al. (2024), Gatscha, Karambelkar & Schloerke (2024), Hijmans (2024a)	
Module 4	Correlation layers	Compute the Pearson correlation coefficient, generate a correlation matrix, and calculate the Variance Inflation Factor (VIF). Exclude layers with VIF values greater than 10.	usdm	Naimi et al. (2014)	
Module 5	Points and pseudoabsences	Randomly generate pseudo-absence points within the study area.	dismo	Hijmans et al. (2024b)	
Module 6	biomod2	Develop ENM using GLM, GAM, GBM, CTA, ANN, BIOCLIM, SRE, FDA, MARS, RF, MAXENT, MAXNET, and XGBOOST.
Evaluate model performance using ANOVA, AIC, AUC, Cohen’s Kappa, and TSS.
Generate response curves, results, metrics, and maps in .tiff format.	biomod2, MIAmaxent	Vollering, Halvorsen & Mazzoni (2019), Thuiller et al. (2024)	
Module 7	Load and plot maps	View models geographic prediction generated biomod2 Module. Ppload .asc, .tif, or .bil files to display the ENM maps on an interactive interface. The raster map is overlaid on a satellite or OpenStreetMap base layer, allowing clear visualization of the study area and raster values for detailed analysis.	leaflet, leaflet.extras
terra	Hijmans (2024b), Cheng et al. (2024), Gatscha, Karambelkar & Schloerke (2024)	
Module 8	Partial ROC analysis	Evaluate model performance using partial ROC analysis.	ntbox	Robin et al. (2011), Osorio-Olvera et al. (2020)	
Module 9	Remove urbanization	Extract urban areas from the ENM or other layers.	terra, raster	Hijmans (2024a, 2024b)	
Module 10	Calculate area	Determine area suitability in km² for the target species. Use the selected threshold to visualize and plot this suitability area. Load an .asc file and adjust the suitability threshold value as needed.	terra, raster	Hijmans (2024a, 2024b)	
Module 11	Gains and losses plot	Compare changes in potential distribution maps across different timeframes or models.	terra	Hijmans (2024b)	
Module 12	Niche overlap analysis via ENMTools	Compare landscape conditions and quantify similarity between ENM using Schoener’s D, Hellinger’s I, env D, env I, and envcor, along with linear, blob, and ribbon range-break tests. Additionally, compare habitat suitability estimates and perform equivalence and similarity tests.	ENMTools	Warren & Dinnage (2024)	
Module 13	Functional landscape connectivity	Invert the ENM output to generate a resistance map. It then allows user to assess functionally connectivity using circuit theory, or the least-cost path approach.	sf	Pebesma & Bivand (2023)	

Figure 3 Directory structure of the EcoNicheS application.

The main folders and the resulting outputs generated after completing the full workflow.

Module 1: environmental data

This module enables users to download and process the 19 bioclimatic layers from WorldClim using the geodata (Hijmans et al., 2024a) and terra (Hijmans et al., 2024b) R packages, with options to obtain global or country-specific data. Moreover, monthly climate variables, including minimum and maximum temperature (Tmin, Tmax, respectively), precipitation, wind, and vapor pressure (vapr), are available. Users can select among three spatial resolutions (10°, 5°, 2.5°, and 0.5°) or upload custom environmental layers. The downloaded data can be clipped using an interactive map, by country, or by uploading a shapefile (.shp) or ASCII (.asc) mask. The final clipped layers are stored in .asc format (Sunny et al., 2024a).

Module 2: get and clean GBIF data (Occurrence processing)

The first step in niche modeling is obtaining species occurrence records. EcoNicheS provides two options: downloading records from the Global Biodiversity Information Facility (GBIF) via the rgbif (Chamberlain et al., 2024) R package or by integrating user-supplied data in a comma-delimited text file (.csv). Because ENM predictions can be biased by museum collections, accessibility-related sampling biases, and taxonomic misidentifications (Araújo & Guisan, 2006; Boria et al., 2014; Sillero & Barbosa, 2021), EcoNicheS applies data-cleaning procedures using CoordinateCleaner (Zizka et al., 2019) package. This process removes erroneous records, such as those located at country or province centroids, open oceans, or those containing atypical or invalid coordinates. Additionally, duplicate records are filtered by selecting one occurrence per environmental pixel (e.g., 1 km2). The spThin package (Aiello-Lammens et al., 2015) package is also implemented to reduce spatial bias by thinning occurrence in over-sample regions. Users can visualize records before and after cleaning on an interactive map. The final cleaned dataset is downloaded in .csv format for use in Module 5: Points and Pseudo-Absences (Sunny et al., 2024a).

Module 3: load and plot maps

This visualization module allows users to upload .asc environmental layers and display them using the raster (Hijmans, 2024a) R package. Users can save maps as .pdf files, zoom into areas of interest, and analyze raster values interactively using the leaflet (Cheng et al., 2024) and leaflet.extras (Gatscha, Karambelkar & Schloerke, 2024) R packages. This module is particularly useful for inspecting input layers before ENM modeling (Sunny et al., 2024a).

Module 4: correlation layers

Collinearity among environmental variables can cause overfitting and reduce model interpretability significantly (Dormann et al., 2013, Graham, 2003; Feng et al., 2019). Therefore, assessing variable collinearity is crucial for reducing the number of variables, enhancing model efficiency, and simplifying interpretation. Various strategies exist to evaluate the acceptable degree of correlation between variables. In this module, users can set a correlation threshold (0–1) by performing Pearson correlation analysis, generating a heatmap to visualize collinearity between layers (Sunny et al., 2024b). The module also calculates the variance inflation factor (VIF) using the usdm (Naimi et al., 2014) R package. Variables with VIF >10 are excluded to improve model efficiency (Sunny et al., 2024a).

Module 5: points and pseudoabsence points

In ecological niche modeling, pseudoabsences are generated to sample the environmental space of a region (Phillips et al., 2009; Whitford, Shipley & McGuire, 2024). Various methods exist for generating pseudoabsences, with one of the most common approaches involving random sampling across the calibration area (Araújo et al., 2019; Valavi et al., 2022). This ensures that broad range environmental conditions are represented (Valavi et al., 2022). The EcoNicheS platform facilitates pseudoabsence generation using the dismo (Hijmans et al., 2024b) package, allowing users to create points that are randomly distributed within a user-defined geographical area. Selecting an adequate number of pseudoabsence points is crucial to ensure a comprehensive environmental representation while avoiding model biases (Merow, Smith & Silander, 2013; Franklin, 2023). However, users should be mindful that a higher number of pseudoabsence points increases computational demands (Valavi et al., 2022; Sunny et al., 2024a).

The module output is a comma-separated values (.csv) file, which can be further processed in other R packages or spreadsheet applications (Fig. 3). The dataset is formatted for direct integration into Module 6: biomod2 (ENM), where it can be used in biomod2 (Thuiller et al., 2024) package. To facilitate visualization, the module initially generates a basic map displaying the spatial distribution of the generated pseudoabsences, which can be downloaded as a PDF. Additionally, an interactive map allows users to visually inspect the stored dataset in their working directory, displaying both the original presence points and the newly generated pseudoabsence points. Within the .csv database, presence and pseudoabsence points are labeled with a “response” value: 1 denotes presence points, while 0 represents pseudoabsence points. Before running analyses, the dataset name is automatically updated while retaining the .csv extension, ensuring seamless integration and proper file saving (Sunny et al., 2024a).

Module 6: biomod2 (ENM)

A wide range of algorithms are available for ENM, each using different methodologies. Ensemble models, which combine multiple algorithms, enhance predictive accuracy by compensating for the limitations of individual approaches. EcoNicheS facilitates the creation, calibration, and evaluation of these ensemble models through an intuitive interface. The plataform employs the biomod2 R package, a widely used ecological niche modeling framework that supports 12 correlative algorithms: generalized linear models (GLMs), generalized additive models (GAMs), generalized boosting models (often called boosted regression trees; GBM), classification tree analysis (CTA), artificial neural networks (ANNs), surface range envelopes or BIOCLIM (SRE), flexible discriminant analysis (FDA), multiple adaptive regression splines (MARS), random forest (RF), and maximum entropy (MAXENT) using the MIAmaxent (Vollering, Halvorsen & Mazzoni, 2019) R package MAXNET and extreme gradient boosting (XGBOOST) (Huang et al., 2023). Users can select multiple algorithms based on their specific research objectives and data availability, as each method has distinct strengths and limitations (Sunny et al., 2024a).

EcoNicheS also incorporates model evaluation metrics available in biomod2, offering two main types of analysis of assessment: goodness-of-fit (e.g., ANOVA, AIC) and model accuracy, which includes metrics such as the Area Under the Curve (AUC), Cohen’s Kappa, and the True Skill Statistic (TSS). The AUC measures a model’s ability to discriminate between presence and absence across all probability thresholds, with values ranging from 0.5 (random prediction) to 1.0 (perfect discrimination). Its simplicity and ability to summarize model performance across different thresholds make it a widely used metric (Phillips, Anderson & Schapire, 2006). The TSS is defined as {1 − the maximum value of (sensitivity + specificity)}, where sensitivity and specificity are calculated based on the probability threshold that maximizes their sum. TSS values range from −1 to +1, with values close to +1 indicating perfect agreement between the observations and predictions, while values of 0 or lower suggest performance no better than random (Allouche, Tsoar & Kadmon, 2006; Franklin, 2010). Similarly, Kappa statistic also ranges from −1 to +1, with +1 indicating high prediction consistency and values near 0 or lower suggesting performance equivalent to random chance (Thuiller et al., 2009). At the end of the modeling process, users can generate graphical outputs, including response curves, evaluation metrics, and prediction maps. Additionally, the module exports model outputs as .tiff files, which are automatically stored in a designated folder for easy access and integration into subsequent workflow steps (Fig. 3) (Sunny et al., 2024a).

Module 7: load and plot maps

In this module, you can upload .asc, .tiff, .tif, or .bil files to display the ecological niche projected on to a map on an interactive interface. The map is overlaid on a satellite or OpenStreetMap (OSM) base layer, allowing visualization of the study area with the raster values clearly shown for detailed analysis. Additionally, this module includes a zoom function, allowing users to focus on specific areas of interest for a more detailed examination of the map (Sunny et al., 2024a).

Module 8: partial ROC analysis

The AUC has long been the gold standard for evaluating model performance in ecological niche modeling, primarily due to its intuitive interpretation, indicating the probability that the model ranks a random presence higher than a random absence. However, when reliable absence data are unavailable, the probabilistic interpretation AUC values becomes problematic, requiring a reconsideration of model comparisons methods (Jiménez & Soberón, 2020).

To address this limitation, partial receiver operating characteristic (pROC) analysis, a threshold-independent evaluation proposed by Peterson, Papeş & Soberón (2008), focuses on specific subsectors of the ROC space by setting an acceptable true positive error threshold. The pROC method evaluates the relationship between the omission error for independent points and the proportion of the area predicted to be suitable for the species. However, this approach is valid only under low omission error conditions. The resulting AUC ratios (calculated as the partial AUC divided by random expectations) range from 0 to 2, where a value of one indicates random performance (Peterson & Soberón, 2012; Peterson, Papeş & Soberón, 2008). This method enhances model evaluation by mitigating artificially low AUC values, particularly in presence-only datasets, where conventional ROC analysis and AUC interpretation can be misleading (Sunny et al., 2024a).

EcoNicheS facilitates model performance assessment by calculating the partial ROC curve using the pROC (Robin et al., 2011) and ntbox (Osorio-Olvera et al., 2020) packages. This module allows users to input a raster of continuous values and compare them with species presence points (Sunny et al., 2024a), following the method proposed by Peterson, Papeş & Soberón (2008).

Module 9: remove urbanization

Urbanized areas represent heavily modified environments that may not accurately reflect the ecological processes driving species distributions (Kondratyeva et al., 2020). Excluding these areas allows EcoNicheS to focuses on natural habitats, improving the quantification of suitable areas. This module enables users to remove urbanized region or other landscape features from the species distribution model using a raster layer containing urban coverage data for the region of interest. The process is conducted using the raster package. Additionally, users can visualize the modified distribution map and export it in .asc format for further analysis (Sunny et al., 2024a).

Module 10: calculate the area

This module calculates the total area of suitability (in km2) for a given species using the raster package in R, this function estimates the surface area of cells in an unprojected longitude/latitude raster, using an approximation based on the cell’s latitudinal height and its longitudinal width, calculated at the cell’s midpoint (Hijmans, 2024b). Therefore, user must be aware that area approximation is less precise at higher latitudes, with the greatest variation near the poles. Users can upload the ENM file generated in Module 8 or provide an external raster file for analysis. The calculation is based on continuous suitability map ranging from 0 to 1. Once the raster file has been loaded, users can define a suitability threshold to filter cells. Cells with values below the selected threshold are excluded from the analysis, as they are assigned a “not available” (NA) value. The area function in the raster package is then used to calculated the area of each remaining valid cell. Subsequently, the total area of suitability is determined by multiplying the number of valid cells by the average cell size. The final result is presented as the total suitable area in square kilometer (km2). By default, EcoNicheS uses the EPSG:4326-WGS 84 geographic projections consistent with the coordinate system used in the ENM generate with biomod2 (Sunny et al., 2024a).

Module 11: gains and losses plot

The module enables users to visualize areas of change using the raster package, allowing analysis of potential gains or losses in species distribution between two time points. This module requires two raster files in .tif or .asc format. The first file represents the current data or environmental characteristics of the area of interest, while the second file contains projections for future or past environmental conditions. Users can also download the generated maps for further use (Sunny et al., 2024a).

Module 12: niche overlap analysis via ENMTools

Niche overlap occurs when coexisting species share portions of their ecological niche. According to ecological and evolutionary displacement models, species should exhibit low niche overlap due to competition, promoting resource partitioning and biodiversity (Pianka, 1974; Schoener, 1974). EcoNicheS quantifies niche similarity using the ENMTools package (Warren & Dinnage, 2024), applying multiple niche overlap metrics to compare species ENM.

The module employs two widely used overlap indices: Schoener’s (1968) and Hellinger’s distance-based metric ‘I’ (Warren, Glor & Turelli, 2008). Additionally, it incorporates new metrics, including env.I, env.D, and env.cor (Warren et al., 2021), which assess niche similarity in the n-dimensional space rather than restricted comparison to observed environmental conditions within the training region. These metrics compare suitability estimates across grid cells using, normalizing suitability scores so that they sum to 1 within the geographic space.

Beyond niche overlap indices, ENMTools provides other similarity measures, such as Spearman’s rank correlation coefficient, and enables hypothesis testing of niche equivalence and similarity using rangebreak tests (Glor & Warren, 2011). These tests help assess whether species ecological niche and its distribution are shaped by environmental barriers or random processes. For example, the Linear and blob tests evaluate whether the geographic regions occupied by two species are more environmentally distinct than expected by chance. The ribbon test is designed to test whether the ranges of two species are separated by an unsuitable habitat barrier, potentially indicating biogeographic process such as speciation or environmental constraints (Glor & Warren, 2011; Sunny et al., 2024a).

Module 13: functional connectivity

Landscape connectivity refers to the degree to which an environment facilitates or restricts organism movement between different locations (Taylor et al., 1993; Tischendorf & Fahrig, 2000). EcoNicheS includes a specialized functional connectivity module that integrates three key tools: Map Inverter, Connectivity Circuit Theory, and Least-Cost Path (LCP) Corridors. These tools generate flow maps and identify optimal movement routes using the gdistance (van Etten, 2017) R package. The analyses require a resistance raster, which represents movement difficulty, and species occurrence data in .csv format. Users can upload custom resistance layers optimized for their specific study context, as resistance values can be derived using various methods (Peterman, 2018).

By default, EcoNicheS derives resistance layers by inverting suitability maps generated through biomod2. The Map Inverter tool transforms suitability predictions into resistance values, under the assumption that areas with higher suitability offer lower resistance to movement. This approach has been applied in studies such as Martinez-Martinez et al. (2024), Rubio-Blanco et al. (2024), and Ruiz-Reyes et al. (2024).

The computation of flow maps and LCP is based on graph theory, modeling resistance surfaces as graphs. The Circuit Connectivity Theory tool converts resistance layer into graphs, where cell centers act as nodes connected by edges weighted by resistance, cost, or friction values, reflecting the difficulty of movement. Alternatively, conductance (1/resistance) can be used to evaluate landscape permeability, offering flexible assessments of connectivity in complex, heterogeneous landscapes (van Etten, 2017).

The LCP Corridors tool identifies the most efficient movement routes between two points, minimizing total movement costs across the resistance surface. A transition matrix, similar to that used in flow map computation, represents movement costs between cells. The algorithm then determines the path with the lowest cumulative cost, modeling potential corridors for species movement (Sunny et al., 2024a).

This module generates multiple outputs for detailed connectivity analysis. The main output is a flow map raster layer in .asc format, which can be used for GIS-based visualization and further editing. For LCP analysis, the module generates a node distance file that records distances between connected nodes, a file containing corridor and LCP values, and an LCP raster file in .asc format, providing a spatial representation of movement corridors. Together, these tools offer a comprehensive framework for visualizing and quantifying connectivity patterns, supporting applications in ecological, conservation, and landscape planning studies (Sunny et al., 2024a).

Empirical example: the distribution and connectivity of Tapirus bairdii in the Selva Maya

To demonstrate the application of EcoNicheS, we conducted a case study on the Central American tapir (Tapirus bairdii), a threatened Neotropical mammal facing severe habitat loss and fragmentation due to land-use changes in the Selva Maya (Falconi-Briones et al., 2025). The Selva Maya, a tropical rainforest conservation initiative spanning Mexico, Guatemala, and Belize, is a priority area for biodiversity conservation (Laako et al., 2022). We modeled the T. bairdii niche and projected onto this region, demonstrating the capabilities of EcoNicheS in processing bioclimatic layers and species occurrences, also developing ecological niche model, and evaluating its performance. Additionally, we analyzed the ecological niche overlap between T. bairdii and the white-lipped peccari (Tayassu pecari), a species that has also suffered severe population declines due to overhunting and deforestation in the Selva Maya (Reyna-Hurtado, Rojas-Flores & Tanner, 2009; Falconi-Briones et al., 2022; Sunny et al., 2024a).

To construct both niche models, we gathered occurrence data from the GBIF and complement it with our own records. The dataset was cleaned using the “Clean My Own database” module of EcoNicheS, where occurrences located within 5 km of each other were removed to reduce spatial bias (Sunny et al., 2024a). This process resulted in a total of 356 occurrence points, aligning with the movement range of the Central American tapir in the region (Reyna-Hurtado et al., 2012; Rivero et al., 2022). We then analyzed correlations among the environmental layers using the “Correlation layers” module, obtaining Pearson correlation values and the variance inflation factor (VIF). Environmental layers with VIF > 8 were excluded, resulting in the selection of seven predictor variables: bio3 (isothermality), bio4 (temperature seasonality), bio6 (minimum temperature of the coldest month), bio7 (annual temperature range), bio9 (mean temperature of the driest quarter), bio15 (precipitation seasonality), and bio18 (precipitation of the warmest quarter) (Sunny et al., 2024a). We then generated 100,000 pseudoabsences points using the “Points and Pseudoabsences” module to improve coverage of the study area (Valavi et al., 2022; Sunny et al., 2024a).

The ecological niche model and potential distribution was developed using the biomod2 module. Model calibration was conducted using the block cross-validation strategy using 80% of the occurrence data for training and 20% for testing (Sunny et al., 2024a). Validation metrics included Kappa, TSS, and ROC. The modeling process involved 10 repetitions, applying a model selection threshold of TSS equal to 0.4 (Fig. 4A) The ensemble model combined three algorthims: Maxent, GLM, and RF (Figs. 4B–4D). We also explored the niche overlap between T. bairdii and Tayassu pecari using the “Niche Overlap Analysis” module (Fig. 5). Hypothesis tests for niche identity or equivalence were performed following Warren et al. (2021), including three similarity test (niche identity, symmetric, and asymmetric similarity) and the rangebreak test (Fig. 6). These tests were conducted using Maxent with four replicates (Sunny et al., 2024a).

Figure 4 Evaluation and visualization of species distribution modeling for Tapirus bairdii.

(A) Performance comparison of the models, highlighting the area under the curve (AUC) and Kappa statistics. (B) Boxplot depicting overall model accuracy. (C) Ensemble distribution model showing the probability of environmental suitability for T. bairdii across the Selva Maya region. (D) Binary distribution model illustrating the species’ predicted presence or absence.

Figure 5 Niche identity analysis of Tapirus bairdii and Tayassu pecari.

(A) Niche identity test, with sp1 denoting T. bairdii and sp2 denoting T. pecari. (B) Asymmetric background similarity hypothesis test. (C) Symmetric background similarity hypothesis test.

Figure 6 Ribbon range-break tests indicating whether the distributions of two species are separated by an area unsuitable for one or both species.

In addition to species ecological niche modeling and its distribution, we assessed the functional connectivity of the Central American tapir (Fig. 7) using the “Functional Connectivity” module (Sunny et al., 2024a). For this analysis, we used the previously generated biomod2 model and transformed its output raster using the MapInverter tool in EcoNicheS. The resulting connectivity raster was processed in QGIS to visualize the corridors and linkages between protected areas for the Central American lowland tapir within the Selva Maya.

Figure 7 (A and B) Ecological connectivity map.

The functional connectivity (Circuit Theory) generated by the ecological connectivity module, along with the locations of protected areas (PNAs). It highlights how these areas affect the connectivity of Tapirus bairdii populations in the Selva Maya region.

This case study highlights the versatility of EcoNicheS for ecological niche modeling and connectivity analysis (Sunny et al., 2024a). Additional applications of the platform are demonstrated in several studies conducted during its early development stages (Sunny et al., 2023; Martinez-Martinez et al., 2024; Rubio-Blanco et al., 2024; Sunny et al., 2024a).

Discussion

A fundamental principle behind the development of EcoNicheS is to provide an accessible, user-friendly platform for ecological niche modeling, eliminating the need for users to build models from scratch (DeAngelis et al., 2021; Sunny et al., 2024a). By leveraging the ShinyDashboard framework, EcoNicheS streamlines complex modeling workflows through an intuitive interface, reducing reliance on programming expertise in R—an essential requirement in many ENM analyses. This accessibility benefits researchers and conservation professionals who require robust modeling tools but may face technical barriers that have traditionally limited the adoption of ENM methodologies (DeAngelis et al., 2021; Sunny et al., 2024a).

EcoNicheS is a comprehensive platform designed for species ecology, biodiversity conservation, and biogeography, integrating a wide range of analytical tools to enhance the accuracy, reliability, and reproducibility of ecological niche models (Feng et al., 2019; Sunny et al., 2024a). By consolidating niche modeling tasks within a single, cohesive interface, EcoNicheS reduces the need for fragmented workflows across multiple software environments. The platform streamlines essential modeling processes, including species occurrence data processing, model performance evaluation, and visualization, while employing ensemble modeling techniques to improve predictive accuracy and mitigate individual algorithm weaknesses (Sunny et al., 2024a).

Beyond technical capabilities, EcoNicheS simplifies access to advanced ENM techniques, allowing users—regardless of programming expertise—to generate replicable and high-quality models while adhering to best practices recommended by experts in the field (Araújo & Guisan, 2006; Sillero & Barbosa, 2021). Additionally, it addresses critical ENM challenges such as managing data bias, selecting appropriate environmental variables, and ensuring robust model predictions, making it a valuable tool for researchers and conservation professionals alike.

Sampling bias is a significant issue in species occurrence records, complicating the development of accurate ENMs (Cuervo et al., 2023; Sunny et al., 2024a). EcoNicheS tackles data quality issues through the integration of the CoordinateCleaner package, which automatically filters out erroneous records from online databases, reducing the time required for manual data cleaning (Zizka et al., 2019; Sunny et al., 2024a). To further mitigate sampling bias, EcoNicheS incorporates the SpThin package, which thins species presence records in oversampled regions while preserving the maximum amount of valuable information (Aiello-Lammens et al., 2015; Sunny et al., 2024a). Another common issue in ENMs is collinearity among predictor variables, which can reduce model efficiency and increases uncertainty, leading to inflated variance, biased estimates, and challenges in interpreting variable effects (Dormann et al., 2013; De Marco & Nóbrega, 2018). EcoNicheS addresses this problem by incorporating a correlation assessment module, allowing users to identify and remove multicollinear variables by calculating the Pearson correlation coefficient and variance inflation factor (VIF) (Sunny et al., 2024a). This ensures that each variable contributes meaningfully to the model, minimizing redundancy and overfitting.

Species distribution models are crucial for predicting future species distributions based on current ecological niche constraints, particularly in the context of climate change. Understanding how suitability may shift under future climate scenarios is a key objective of ENM (Wiens et al., 2019; Araújo et al., 2019; Sunny et al., 2024a). EcoNicheS facilitates research on potential distribution changes, enabling the identification of priority areas for conservation and informing management strategies to mitigate biodiversity threats in a rapidly changing world. The integration of ENM with climate models strengthens biogeographic hypotheses and supports data-driven decision-making by conservationists and resource managers (Wiens et al., 2009; Sunny et al., 2024a). The platform supports multiple modeling techniques through the biomod2 package, enabling users to select the most appropriate methods for accessing their data. A key feature of biomod2 is its ability to combine multiple predictions into a single-ensemble model. In EcoNicheS, this functionality allows the integration of various modeling algorithms, along with three statistical evaluation metrics: the ROC-based AUC, Cohen’s Kappa, and the TSS. The integration of biomod2 tools (Thuiller et al., 2009, 2024; Sunny et al., 2024a) provides access to robust techniques for model development and validation (Sillero et al., 2023; Sunny et al., 2024a).

The ensemble modeling capabilities of biomod2 through EcoNicheS significantly enhance predictive accuracy, making it an indispensable tool for biodiversity conservation and ecological hypothesis testing (Sunny et al., 2024a). Additionally, EcoNicheS incorporates the partial receiver operating characteristic (partial-ROC) method to evaluate model performance, particularly addressing issues related to low-performing algorithms and high omission rates (De Marco & Nóbrega, 2018; Sunny et al., 2024a). At the conclusion of the modeling process, partial-ROC analysis (Peterson, Papeş & Soberón, 2008; Osorio-Olvera et al., 2020; Sunny et al., 2024a) provides a more nuanced assessment of model accuracy, ensuring that model predictions remain statistically robust and ecologically meaningful (Sunny et al., 2024a).

Comparison with similar tools

EcoNicheS provides a comprehensive open-access suite of tools that integrate advanced modeling, niche overlap analysis, and ecological connectivity functions, setting it apart from other similar applications (Table 2). For instance, while the Niche Toolbox (Osorio-Olvera et al., 2020) supports niche analysis, it lacks features such as pseudoabsence generation or ensemble modeling; the complete suite of biomod2 analyses, interactive map visualization; urbanization filters; and ecological connectivity tools (Sunny et al., 2024a). These functionalities are essential for refining ecological niche models and improving their application in conservation planning. However, Niche Toolbox offers a notable feature—the calculation of minimum volume ellipsoids between modeling algorithms, providing an alternative approach for niche definition.

Table 2 Comparison of analyses performed by different software packages for ecological niche modeling.

Analyses	EcoNicheS	Wallace	ENMTML	SDMapp	Niche Toolbox	sASSDM Toolbox	
Platform	R/Shiny dashboard for ENM	R/Shiny dashboard for ENM	R package with Shiny interface	R/Shiny dashboard for ENM	Shiny for ENM	Python based/ArcGIS	
Occurrence data
processing	GBIF API integration and user-provided CSV, spatial filtering with SpThin, data cleaning with CoordinateCleaner	GBIF data import, data cleaning through built-in methods	No direct GBIF integration; user-provided data; background point generation	User-provided data; automatic download from GBIF	GBIF data integration, custom upload, filtering, and bias correction	Custom CSV, shapefiles, rarefaction	
Environmental
data	19 bioclimatic layers from WorldClim, monthly variables (Tmin, Tmax, Precipitation), user-uploaded layers	19 bioclimatic layers from WorldClim, customizable layers	19 WorldClim variables, optional user-provided layers	19 bioclimatic variables from WorldClim or other climate datasets	WorldClim variables, user-uploaded layers, dynamic environmental layers from multiple sources	19 bioclimatic layers from WorldClim, monthly variables (Tmin, Tmax, Precipitation),
user-uploaded layers	
Layer correlation
analysis	Pearson correlation matrix, Variance Inflation Factor (VIF) and Heatmap	Pearson correlation matrix, manual selection	Not available	Variance Inflation Factor (VIF) implemented	Pearson correlation matrix, user-defined thresholds for variable selection	Not available	
Pseudoabsence/Background points	Randomly generated Pseudoabsence points, export to CSV	Generates pseudo-absence/background points via Maxent or random	Automated pseudoabsence points generation	User-defined pseudo-absence/background generation	Automated generation of background points or pseudoabsences	Gaussian Kernel Density of Sampling Localities, Sample by Buffered Local Adaptive Convex-Hull, Sample by Buffered MCP, background selection using bias files	
Modeling algorithms	Integration with biomod2, includes GLM, GAM, GBM, CTA, ANN, BIOCLIM, SRE, FDA, MARS, RF, MAXENT, MAXNET, XGBOOST.	Includes Maxent, Bioclim, Random Forest, BRT, Domain	Maxent, SVM, GLM, RF, GAM, GARP, BioClim	Maxent, Random Forest, Boosted Regression Trees (BRT)	Minimum volume ellipsoids, BIOCLIM, Maxent	Maxent	
Ensemble modeling	Available through biomod2, combines multiple algorithms	Not available	Available	Not available	Not available	Not available	
Model evaluation	ROC, AUC, Partial ROC, TSS, Kappa, Boyce Index.	ROC, AUC, TSS, Kappa	ROC, AUC, TSS, Kappa	ROC, AUC, TSS	ROC, AUC, Partial ROC, Boyce Index, Confussion matrix metrics	ROC, AUC	
Map visualization	Interactive map visualization maps in .asc format, save in .pdf	Map viewer for distribution maps	.asc file visualization	Real-time map visualization with .asc export	Interactive map visualization and downloading in .tiff, .asc	Raster file, GIS visualization	
Urbanization filter	Module for removing urbanized areas from models	Not available	Not available	Not available	Not available	Not available	
Area calculation	Calculate suitable area in km2 for species	Area calculation based on occurrence points	Area calculations based on presence	Basic area calculations	Area calculations based on habitat suitability thresholds	Calculate suitable area in km2	
Gains and losses analysis	Comparison of present and future distribution maps	Not available	Not available	Not available	Not available	Not available	
Niche overlap analysis	Uses ENMTools for calculating niche overlap, Schoener’s D, Hellinger’s I, env D, env I and envcor, and linear, blob, and ribbon rangebreak tests	Niche overlap Schoener’s D, Hellinger’s I with ecospat	Not available	Not available	Not available	Not available	
Ecological connectivity analysis	Flow maps using gdistance	Not available	Not available	Not available	Not available	Not available	
Output formats	CSV, .tiff, .asc, .pdf	CSV, .tiff, .pdf	CSV, .tiff	CSV, .tiff	CSV, .tiff, .asc, .pdf	CSV, .tiff, .asc	

Various ecological niche modeling platforms offer different strengths and limitations. Wallace 2 (Kass et al., 2023) and SDMapp (Hema et al., 2021) focus primarily on niche modeling via Maxent, limiting flexibility for multi-algorithm ensemble modeling (Sunny et al., 2024a). ENMTML (de Andrade, Velazco & De Marco Júnior, 2020) provides a broad range of algorithms but lacks spatial analysis tools, while SDMToolbox 2.0 (Brown, Bennett & French, 2017) excels in GIS-based connectivity and least-cost path analyses, though its reliance on ArcGIS reduces accessibility compared to open-source alternatives. EcoNicheS integrates a broader set of functionalities, including urbanization filters, advanced niche overlap analysis, and spatial modeling, ensuring a more comprehensive workflow for species distribution modeling and connectivity assessments (Table 2).

While no single platform meets all analytical needs, integrating multiple tools can enhance reproducibility and analytical depth in ENM research (Kass et al., 2024). EcoNicheS consolidates a diverse suite of modeling approaches but complementing it with specialized platforms for specific tasks—such as minimum volume ellipsoid modeling (Niche Toolbox) or detailed GIS analyses (SDMToolbox)—can provide additional insights. This flexibility in combining multiple modeling techniques is essential for addressing complex ecological and conservation questions (Sunny et al., 2024a).

Although EcoNicheS enhances accessibility through a user-friendly interface, successful application of ENM still requires a strong theoretical foundation in niche theory, ecological principles, and statistical methods. Users must make informed decisions regarding predictor selection, bias correction, and model interpretation to ensure ecologically meaningful results. While streamlined workflows improve usability, over-reliance on automated tools without scientific rigor can lead to misinterpretation (Elith & Leathwick, 2009; Araújo & Peterson, 2012). Achieving a balance between accessibility and methodological soundness is essential for ensuring robust, reproducible, and actionable conservation outcomes.

Empirical case

A case study of the Central American tapir (Tapirus bairdii) in the Selva Maya demonstrated the practical application of EcoNicheS (Sunny et al., 2024a). The workflow, which was aligned with best practices in data cleaning, thinning, modeling, and evaluation, was followed. Results from biomod2 indicated that the random forest (RF) algorithm outperformed Maxent in terms of statistical performance and accuracy (Figs. 4A, 4B; Sunny et al., 2024a). The final ensemble model highlighted greater environmental suitability for T. bairdii in southeastern Selva Maya, particularly between the protected areas of Belize and northern Guatemala (Fig. 4C), covering a total area of 98,282.1 km2 (Fig. 4D) (Sunny et al., 2024a).

Niche overlap analysis between T. bairdii and Tayassu pecari (Fig. 5) provided no evidence to reject the null hypothesis that their niches are identical, with Schoener’s D = 0.2 and Hellinger’s I = 0.2. Environmental variable correlation tests confirmed this, showing no significant differences (Env D = 1.0, Env I = 1.0) and consistent results in symmetric and asymmetric background tests (Figs. 5B, 5C) (Sunny et al., 2024a).

Connectivity analysis using gdistance (van Etten, 2017) highlighted key corridors and least-cost paths (LCPs) linking protected areas in Belize and northern Guatemala, indicating high connectivity potential for tapirs (Fig. 7). In contrast, lower connectivity was observed among tapir populations in Mexico, particularly in the southern Yucatán Peninsula (Fig. 7B), underscoring the need for targeted conservation strategies in this region (Sunny et al., 2024a).

Conclusion

EcoNicheS offers a powerful and accessible platform for conducting ecological niche modeling, incorporating various algorithms, visualizations, and functionalities (Sunny et al., 2024a). The package empowers researchers and conservation practitioners to perform comprehensive and replicable ENMs analyses. As ecological challenges continue to evolve, tools such as EcoNicheS will play an increasingly important role in guiding biodiversity conservation efforts and fostering innovation in ecological research. Furthermore, EcoNicheS will continue to be updated to introduce new analyses and functionalities that address emerging research needs (Sunny et al., 2024a).

For a tutorial on how to use EcoNicheS, a comprehensive video guide is available on YouTube at https://www.youtube.com/watch?v=fLul1kkZhfI&t=26s. Additional information, including installation instructions, user documentation, and updates, can be found on the official EcoNicheS website: https://armandosunny.github.io/EcoNicheS/. For technical support, inquiries regarding software functionality, or to report any issues, users are encouraged to contact the development team via email at econichesapp@gmail.com.

Supplemental Information

Supplemental Information 1 EcoNicheS R code.

We are grateful to the editor and four anonymous reviewers for their comments. Portions of this manuscript were previously published as part of a preprint (Sunny et al., 2024a). The authors acknowledge the contributions of all co-authors and reviewers involved in the development of the original preprint. The preprint is available at Research Square: (https://assets-eu.researchsquare.com/files/rs-5096850/v1/7101d5c6-51c3-46e7-93c1-35fb623db5eb.pdf?c=1726646665). A.S: Adahy Olun Contreras-García, te extraño muchísimo y estoy muy orgulloso de ti. Sigo luchando por volver a estar contigo, mi querido hijo.

Additional Information and Declarations

Competing Interests

Armando Sunny is an Academic Editor for PeerJ.

Author Contributions

Armando Sunny conceived and designed the experiments, performed the experiments, analyzed the data, prepared figures and/or tables, authored or reviewed drafts of the article, and approved the final draft.

Clere Marmolejo conceived and designed the experiments, performed the experiments, analyzed the data, authored or reviewed drafts of the article, and approved the final draft.

Rodrigo Vidal-López performed the experiments, analyzed the data, authored or reviewed drafts of the article, and approved the final draft.

Fredy A. Falconi-Briones analyzed the data, authored or reviewed drafts of the article, and approved the final draft.

Ángela P. Cuervo-Robayo conceived and designed the experiments, performed the experiments, analyzed the data, prepared figures and/or tables, authored or reviewed drafts of the article, and approved the final draft.

René Bolom-Huet conceived and designed the experiments, performed the experiments, analyzed the data, prepared figures and/or tables, authored or reviewed drafts of the article, and approved the final draft.

Data Availability

The following information was supplied regarding data availability:

The code and package is available at GitHub and Zenodo:

- https://github.com/armandosunny/EcoNicheS.

- Sunny, A., & Marmolejo, C. (2025). EcoNicheS: enhancing ecological niche modeling, niche overlap and connectivity analysis using the shiny dashboard and R package. Zenodo. https://doi.org/10.5281/zenodo.14790014.

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
