# Peer review of "EcoNicheS: enhancing ecological niche modeling, niche overlap and connectivity analysis using the shiny dashboard and R package"

_PeerJ, doi:10.7717/peerj.19136_

## Round 0.1 · original submission · Minor Revisions

I agree with the comments provided by the four reviewers. Please address these in your revision.

·

Basic reporting

Overall, this manuscript is well written and of interest to readers of PeerJ.

Experimental design

I agree with the main objectives, the software’s goals and its presentation. As an author of a similar project, SDMtoolbox (www.sdmtoolbox.org), I understand challenges associated and pitfalls with these types of software. I appreciated the thorough Github tutorial. Remember, the easier you make this for non-experts, the more likely they will use it.
A few things to consider to maximize the likelihood that people use this software and minimize the number of issues they face (and minimize the emails you receive).
1) Consider doing a YouTube video of you running through a basic analyses (e.g. example: https://youtu.be/rsvSqTjxkqQ). Younger researches utilized this medium a lot
2) Get a dedicated email for your R package, seriously. It keeps the booking and issues centralized. It also helps to ensure that things do not get lost.

Validity of the findings

Overall, the software worked well and though I didn’t test it exhaustively, I had no major hiccups. As such, I support the publication of this manuscript and its associated software. Excellent job!

Sincerely,
Jason L. Brown

Additional comments

Minor things/issues:
-Some citations are missing, e,g, that of Baecher, 2024 (cited on line 323), this is not in the citations.
-Also note that Warren 2021 is in the citations twice.
- Please double check all references in text and in the citations. I am not great at finding these types of errors and the existence of two suggests that there are several more.

-Consider adding and discussing SDMtoolbox (Brown et al 2017) to the section ‘Comparison with Similar Tools’, lines 423-433. It is very relevant to the topic:

Brown JL, Bennett JR, French CM (2017). SDMtoolbox 2.0: the next generation Python-based GIS toolkit for landscape genetic, biogeographic and species distribution model analyses. PeerJ

Reviewer 2 ·

Basic reporting

The manuscript titled EcoNicheS: enhancing ecological niche modeling, niche overlap and connectivity analysis using shiny dashboard and R Package” is well-written. The authors have provided a comprehensive background to the problem of wide use and adoption of ecological niche modeling by practitioners. The manuscript includes a comparison of the proposed workflow, EcoNicheS and four other ecological niche modeling workflows published in the last 4-5 years. The tables and figures of the manuscript are helping the readers follow the workflow and visualize the inputs and outputs.

Experimental design

This manuscript matches the Aims and Scope of the journal and addresses a need in the ecological niche modeling field for user-friendly platforms to support wide adoption of this type of modeling by practitioners. The authors provided detailed descriptions of each step of the modeling workflow and included a case study to better illustrate the use of the workflow.

Validity of the findings

This manuscript introduces a new workflow for ecological niche modeling that allows users to run models without requiring coding skills. The manuscript provides enough methodological details to inform the reader about the types of data inputs, analyses ran, and interpretation of workflow outputs.

Additional comments

The manuscript is well written and the workflow proposed is robust. I have just a few suggestions and questions for the authors.
Please use the SDM/ENM acronyms consistently throughout the manuscript. The acronyms are introduced as representing species distribution modeling/ecological niche modeling, but on page 2 of Introduction (and later in the manuscript) the acronyms are used as SDMs/ENMs, which stand for species distribution models/ecological niche modes.
Lines 110-111: This sentence is missing a predicate: “SDM/ENM using ensemble modeling facilitated by the biomod2 package.”
Line 125: since the study region is mentioned here for the first time (“Selva Maya”), add more locational information (Province and Country).
Line 237: is it possible to add omission error to the list of calculated model accuracy metrics (besides AUC, Cohen’s kappa, and TSS)?
Line 278: Provide details about the land use/land cover product used to represent urbanized areas. Can this product be changed by the user?
Line 284: Please provide details about the map projection used to calculate suitability area (in sq km). The rasters need to have an equal area map projection for measuring suitable areas. Equal area projections usually are continent (or smaller region) specific.
Figure 4: the figure caption is incomplete – it misses explanations for panels C and D
Figure 5: the figure caption is incomplete – it misses explanations for panels C and D
Figure 6: explain the meaning of the legend (what does a value of 1 represent)?
Figure 7L the legend includes an acronym “PNAs” – spell out the acronym in the legend or in the figure caption.

·

Basic reporting

The Authors present a package in R to create a species distribution modelling platform through Shiny. The package is impressive: it downloads and cleans the environmental and species data, runs and evaluates the models, and also includes tools for testing niche similarity and landscape connectivity. I congratulate the Authors for this amazing work. This package can facilitate and extend the use of ecological niche models. I have some minor changes to suggest. However, I missed an important point: why do I need to save all the data and upload again them to continue working through the flow of the modules? I was expecting that the data should saved automatically, and uploaded into the next module without the necessity of doing it manually. That would simplify the whole process. In any case, this is a very necessary package in the field of ecological niche modelling. My congratulations to the Authors.

Experimental design

General comments:
Only after visiting the GitHub page, did I realise that the package is not available through a webpage, such as other shiny applications. Why? This will hamper the use by potential users, as they will need to install R and the package. Unfortunately, I was not able to run a model: I have problems in the module for creating pseudo-absences. I did not test the package in Windows.
In fact, the installation took a long time to process in Linux Mint 22. I got several errors and I needed to run the script several times. I managed to install the package through the file EcoNicheS.R and by installing some packages manually. Although it is extremely difficult to avoid problems between packages’ versions, this can discourage people from using the package. This is why putting the package on a permanent webpage is so important.
Why is the raster package used in EcoNicheS? The raster package is retired and is substituted by terra. For example, in L180 it is stated that the user can view data with the raster package. This function can be performed with terra. Similarly, dismo package is going to be substituted briefly by the package predicts, by the same authors.
What is the difference between ecological niche models and species distribution models? As explained by Sillero et al 2021, both terms represent the same.
The introduction is really complete. A very nice summary of the use of ecological niche models.
Why is removing urbanisation necessary? This should be explained in detail. How are urbanisation areas identified? What is the origin of the urbanisation layer? Is this layer provided by the user or by the raster package?
Are the niche overlap analyses limited to Maxent? Using dismo functions, it is possible to apply the analyses to any algorithm.
I know that the study case is only an example to prove that the EcoNicheS works perfectly, but why only four repetitions? The minimum should be 10. Less than that does not provide enough statistical power. What is the model selection threshold? Is it to select the models with a proper performance? For that, it is necessary to have all the thresholds calibrated for the name range. AUC ranges from 0 to 1, while TSS goes from -1 to 1. The ranges are not similar.
I am missing a figure presenting the GUI of EcoNicheS.
Figure 1 is not referred to in the text.
I do not understand the Figure 6. It is more of a conceptual example, right?

Validity of the findings

No comments

Additional comments

Minor comments:
L52: Also in past climates, not only present and future, as indicated in L63 and following. ENMs can predict distributions in any climate period if data are available.
L102: Please, present briefly Shiny. Some readers might not know what it is.
L109: Figure 1? Figure 1 is missing so, the first figure that is referenced in the main text is Figure 2.
L200: Do you refer here to background points? It is important to distinguish between background and pseudoabsence points, as this is still a common error. In fact, in L207, background points are mentioned.
L239: Why are TSS and Kappa presented in detail, but not AUC?
L285: The previous module is module 8 or 6? It should be easier to understand if the text refers to the species model, not only to the asc file.
Module 10: Should be the two rasters needed here the current and the future models? It should be better to refer to the current and the projection models. People can also use the package to model past distributions.
L301: Something is missing after between. The sentence is not complete.
In the text, sometimes the figures are called Figure or Fig., for instance in lines 449 and 451.
Please, indicate that Tapirus bairdii is a mammal.
There are some errors in the references.
For example, the reference Arenas-Castro et al 2021 should correspond to Arenas-Castro & Sillero 2021.
Sillero et al 2021b should be Sillero & Barbosa 2021.
Sillero & Barbosa 2021b should be Sillero & Barbosa 2021.

·

Basic reporting

Sunny & Marmolejo et al. present EcoNicheS, a software for the automated implementation of ecological niche modeling in the R online platform Shiny. The software is presented as an intuitive, user friendly tool to predict species distributions using ecological niche estimates and a series of additional metrics. The tool is similar to other platforms that facilitate data management pre-modeling, but offers figures and summaries as it goes, and include additional post-processing tools. Although I am supportive of the package, the algorithms available are the same used by most of the platforms available, and little is done to address a gap in the field regarding algorithm availability beyond diomod2 and maxnet. I believe this tool will be useful for the community and the article will receive citations. I leave some suggestions to move from version 1 to future versions, but find the current version suitable for publication.

Abstract:
The abstract offers little information from the package per se. Authors missed an important opportunity to inform users about the algorithms in the package, what makes this tool different from others, and how this tool complements tools already available to the community. This information could be extracted from table 2.
The presentation of the tool for conservation purposes neglects a broader audience in the areas of biogeography, evolution, and epidemiology.
The link to the software should be included in the Abstract.

Introduction:
A main issue I found when reading the article is the misuse of the terms ecological niche modeling and species distribution modeling. At times these concepts are used as synonyms but also as different methods. In the current form it is unclear if the package offers niche estimation in environmental space, or if the main features are species distribution measurements post-ENM estimation. I recommend to develop a more detailed theoretical framework to describe what the package does (and does not) following, for example, the definition of niche modeling here "In defense of ‘niche modeling’": https://www.sciencedirect.com/science/article/pii/S0169534712000754
There is an overuse of reviews instead of primary research.
Paragraphs are too long, repetitive and do not guide the reader to the need of another ENM software. For example, in the current version too much space is used to introduce users to the ENM field, followed by an abrupt description of a tool that aims to facilitate the “ecological niche modeling process” (line 100). Instead, the introduction should help describe the ENM process in the package.

GitHub
Links should open new tabs instead of redirecting the current page.

Method:
Lines 198, 220, 300 should be edited. It sounds like a copy and paste line from work done in the past.
The description of the pseudoabsences is good, but for some software, such as Maxent, it could be referred as a background points considering how Maxent contrasts density of background vs occurrence points. It would be nice to have the opportunity to add a sampling bias surface or upload a file of background points exported previously, or use one point per pixel of the study area M for researchers more interested in experimental analyses without computer power limitations.
It is unclear the rational behind selecting the modeling algorithms for this software. That is, some of the methods are analytically redundant, i.e., they generate similar outputs because they operate under similar analytical frameworks and assumptions (with some slight parameter variations). Other methods that generate models from different data manipulation could generate a richer landscape of available models for a more robust evaluation of model responses. For example, it is unclear why ellipsoids, Marble, hypervolume, GARP, etc. were not invoked. In the current form, the package does not offer much innovation regarding algorithm availability.
In module 6, authors present a series of evaluation metrics. Under the current approach to generate background points, all these metrics are prone to flaws. i.e., random points do not offer statistical independence to allow fair, robust model-prediction assessments. More advanced data partitioning approaches could help mitigate these errors. E.g., ENMeval: An R package for conducting spatially independent evaluations and estimating optimal model complexity for Maxent ecological niche models - Muscarella - 2014 - Methods in Ecology and Evolution - https://besjournals.onlinelibrary.wiley.com/doi/10.1111/2041-210X.12261

A main failure of most packages to facilitate the workflow and replicability is the emphasis in the geographic space for the data curation and filtering, and for model design and interpretation. Users generally fail to understand what environments species are using from the environments available based on the study area M. Adding a module for clearer data visualization in environmental space will advance the field and will fill a gap in tools available.
Modul 7 is among the most exciting tools. One error presented in this package and others and in Peterson et al. 2008, however, is that the occurrence and environmental data must be split to have points and areas independent. That is, a robust evaluation should be performed in a M area that does not includes the M area used for calibration; so that models should be projected to the evaluation areas and the partial ROC text should be done in such independent regions with independent points. Including all the areas (areas for model calibration and model evaluation) will generally return artifactual significant values. That is to say, the smaller the evaluation area the higher the challenge for the model to demonstrate it can predict independent points.
Module 8 is interesting. I believe that authors could elaborate more in other applications of this tool beyond removing urban areas from the model calibration region.
Module 12 is very innovative and needed. Well done. However, more details will improve this section considering that other section have more details (e.g., kind of data to be used, format and units of outputs, etc.).

Final questions:
How is the model ensemble performed? Using binarized maps? If not, what is the rational in combining rasters with different units?
Can r scripts or workflow be exported to be included as supplementary materials in publications as a means to increase the replicability of the models, which is much needed in the field?

Experimental design

Acceptable for the models in the study case.

Validity of the findings

I did not have the opportunity to run the package but based in the text and figures, and recent publications, it seems the package has passed the beta version. I am not sure how available the authors will be to address questions from the community (The community is not invited to provide feedback or report errors, and there is no email offered specifically to attend inquiries from users).

Additional comments

The package can help improve replicability of studies, much needed in modern science.

---

## Round 0.2 · accepted · Accept

The manuscript is now ready for publication as the authors addressed all reviewer comments.